# Does the Infectious Status of Aphids Influence Their Preference Towards Healthy, Virus-Infected and Endophytically Colonized Plants?

**DOI:** 10.3390/insects11070435

**Published:** 2020-07-11

**Authors:** Junior Corneille Fingu-Mabola, Clément Martin, Thomas Bawin, François Jean Verheggen, Frédéric Francis

**Affiliations:** 1Entomologie Fonctionnelle et Évolutive, Terra, Gembloux Agro-Bio Tech, Liège-Université–Passage des Déportés 2, 5030 Gembloux, Belgium; cmartin@uliege.be (C.M.); fverheggen@uliege.be (F.J.V.); frederic.francis@uliege.be (F.F.); 2Department of Arctic and Marine Biology, UiT The Arctic University of Norway, Framstredet 39, 2019 Tromsø, Norway; thomas.bawin@uit.no

**Keywords:** biological control, insect–plant–microbe interactions, multitrophic interactions, aphid-borne virus pathosystem, host-finding behavior, endophytic fungus, plant volatile organic compound

## Abstract

Aphids (Hemiptera: Aphididae) cause significant damage and transmit viruses to various crop plants. We aimed to evaluate how the infectious status of aphids influences their interaction with potential hosts. Two aphid (*Myzus persicae* and *Rhopalosiphum padi*) and plant (*Nicotiana tabacum* and *Triticum aestivum*) species were used. The preferences of aphids towards healthy, virus-infected (*Potato Leafroll Virus* (PLRV) and *Barley Yellow Dwarf virus* (BYDV)), and endophytic entomopathogenic fungi (EEPF)-inoculated (*Beauveria bassiana* and *Metarhizium acridum*) plants were investigated in dual-choice tests. The headspace volatiles of the different plant modalities were also sampled and analyzed. Viruliferous and non-viruliferous aphids were more attracted to EEPF-inoculated plants compared to uninoculated plants. However, viruliferous aphids were more attracted to EEPF-inoculated plants compared to virus-infected plants, while non-viruliferous insects exhibited no preference. Fungal-inoculated plants released higher amounts of aldehydes (i.e., heptanal, octanal, nonanal and decanal) compared to other plants, which might explain why viruliferous and non-viruliferous aphids were more abundant in EEPF-inoculated plants. Our study provides an interesting research perspective on how EEPF are involved in behavior of virus vector, depending on the infectious status of the latter.

## 1. Introduction

Aphids are herbivorous, sap-feeding insects that are regarded as crop pests in agricultural and horticultural production systems globally [1]. Aphids contribute major economic losses by causing significant damage to plants and transmitting viruses [1]. More than half of all insect-vectored plant viruses are transmitted by aphids by non-persistent, semi-persistent, or persistent modes [2,3]. The host-finding behavior of aphids is specific, and largely explains their role as important vectors of plant viruses. This behavior is mediated, in most cases, by volatile organic compounds (VOCs) that are continuously released by plants [4,5,6,7]. For example, a synthetic blend of 11 VOCs, at concentrations and ratios designed to mimic potato plants, induced a similar behavioral response to *Myzus persicae* (Sulzer), as a natural plant on the olfactometer [8].

Persistently transmitted viruses induce changes in the volatile blends emitted by the plants that they infect, leading to the attraction of virus-free aphids, which enhances propagation [9,10,11]. In contrast, insects carrying plant viruses are sensitive to the volatilome of healthy plants, and preferentially feed on them [10,12,13,14,15,16]. Thus, the behavior of sap-feeding insects likely differs in relation to their infectious status.

Entomopathogenic fungi (EPF) are biological control agents used against sap-feeding insect pests, such as aphids [17,18]. However, some EPF strains are able to colonize plant tissue [19,20,21,22,23,24,25]. Several teams of scientists have already investigated the impact of these endophytic entomopathogenic fungi (EEPF) on the biology and physiology of colonized plants, to describe the relationship between plant pathogens and insect pests, with a primary focus on aphids [26,27,28,29,30,31]. The elevated biosynthesis of toxic secondary metabolites in plant tissues [28,32,33,34], change to nutritional quality [35,36] and stimulation of the immune system have been reported [29,37]. These parameters promote the defense ability of host plants, thus improving fitness [38]. The volatilome of colonized plants is also impacted by the presence of EEPF [27,35,39,40,41].

Some biocontrol agents (such as macroorganisms) and semiochemicals (such as alarm pheromones) that are used to control aphids enhance the spread of viruses [42,43,44]. Thus, it is important to understand the relationship between an EEPF-inoculated plant and an aphid when carrying and not carrying viruses. Such knowledge would make it possible to establish a link between the presence of EEPF in plant tissues and the behavior of a virus vector likely to spread a given virus. This information could potentially be useful for aphid/virus management.

We carried out a comprehensive study using different types of insect–plant–microbe interactions. Under the context of plant virus transmission, we hypothesized that viruliferous aphids behave differently to non-viruliferous aphids towards healthy, virus infected, and EEPF-inoculated plants. To verify these hypotheses, we investigated how the presence of *Beauveria bassiana* (Vuill.) and *Metarhizium acridum* (Humber) in host plant tissues affected aphid preference compared to healthy and virus-infected plants. We then sampled and analyzed headspace volatiles to determine whether they explained aphid behavior based on the different plant modalities. Two insect–plant–virus systems were investigated. First, the Myzus–Tobacco–*Potato Leafroll Virus* (PLRV) (MTP) system: tobacco (*Nicotiana tabacum* L.) is one of the host plants of the *Potato Leafroll Virus* (PLRV, Family *Luteoviridae*, Genus *Polerovirus*), which is principally transmitted by *M. persicae* (Hemiptera: Aphididae) in a persistent manner [45]. Second, the Rhopalosiphum–Wheat–*Barley Yellow Dwarf Virus* (BYDV) (RWB) system: wheat (*Triticum aestivum* L.) is often infected by the *Barley Yellow Dwarf Virus* (BYDV, Family *Luteoviridae*, Genus *Luteovirus*), which is transmitted efficiently by *R. padi* (Hemiptera: Aphididae) in a persistent manner [10].

## 2. Materials and Methods

### 2.1. Plant Cultures

All tobacco (cv. Xanthii) and wheat (cv. Johnson) seeds were sown in autoclaved potting soil. Tobacco seedlings were individually transplanted at the three-leaf stage to pots (7 × 7 × 7 cm). Wheat seedlings were separately transplanted at the two-leaf stage to straight sample containers (VWR; 70 mm height; 33 mm diameter; 60 mL capacity; custom-drilled with a hole 2 mm in diameter). The seedlings were then kept in a climate chamber at 22 ± 1 °C, 70 ± 10% relative humidity (RH) and 16:8 h (light: dark) photoperiod. Tobacco and wheat plants were used for the insects to multiply on, and to perform the behavioral and volatilome analyses.

### 2.2. Virus and EEPF Cultures

PLRV was provided by the Leibniz Institute DSMZ (Braunschweig, Germany) on a PLRV-infected *Physalis floridana* Rydb. (Solanales: Solanaceae) plant. BYDV-PAV was acquired by the Functional and Evolutionary Entomology Laboratory at Gembloux Agro-Biotech from infected wheat plants. Both virus strains were propagated by allowing *M. persicae* and *R. padi* individuals to feed on the PLRV- and BYDV-infected plants, after which they were transferred to healthy tobacco and wheat plants, respectively. Virus stock cultures were further maintained via insect transmission every 2–3 weeks, and newly infected seedlings were kept separately in net cages under the same conditions.

Two entomopathogenic fungi were used: (1) *Beauveria bassiana* ((Balsamo-Crivelli) Vuillemin) strain GHA isolated from the commercial product Botanigard^®^ (Certis Europe, Bruxelles, Belgium) and (2) *Metarhizium acridum* ((Driver & Milner) JF Bischoff, Rehner & Humber) strain IMI330189 isolated from Green Muscle^®^ biopesticide [46], obtained from Reproductive Biology, Science and Technology Faculty, University Cheikh Anta Diop, Dakar, Senegal. For each product, wettable powder was dissolved in a 0.01% Tween^®^ 80 solution in distilled water. Thirty-five microliters of suspension were transferred to Potato Dextrose Agar (PDA: Sigma-Aldrich, St Louis, MO, USA) supplemented with chloramphenicol (0.05 g·L^−1^), and maintained in darkness in an incubator at 25 ± 1 °C for 3 weeks. Spores were collected by scraping the agar surface with a sterile L-shaped spreader (VWR, Radnor, PA, USA), and were suspended in a 0.01% Tween^®^ 80 solution. The concentration was adjusted to 10^8^ spores·mL^−1^ using a Neubauer hemocytometer cell [47]. The ready-to-use suspensions were stored at −20 °C and were used within 48 h.

### 2.3. Insect Rearing

*Myzus persicae* strain MpCh4 and *R. padi* strain Xu were reared in net cages on tobacco and wheat plants, respectively, and were kept in a climate chamber at 22 ± 1 °C, 70 ± 10% RH and 16:8 h (light: dark) photoperiod. Wheat and tobacco plants at the three-leaf stage were provided every 2 and 3 weeks, respectively.

Non-viruliferous (I-) and viruliferous (I+) aphids were obtained by placing 40 to 50 adults on healthy plants (HP) and virus-infected plants (either with PLRV (VP-1) or BYDV (VP-2)), respectively. Adult aphids were allowed to reproduce for 24 h, and were then eliminated. Nymphs were maintained until adults emerged (within about 1 week).

### 2.4. EEPF-Inoculated and Virus-Infected Plants

Tobacco and wheat plants were treated 7 days before use by spraying their leaves using a cosmetic sprayer from Sinide Plastic Spray Bottles (30 mL) with fine mist (0.35 mm nozzle diameter). Two milliliters of 10^8^ spore·mL^−1^ suspension of either *B. bassiana* (BP) or *M. acridum* (MP) were used per plant. Healthy plants (HP) and virus-infected plants (VP-1 and VP-2) were sham-inoculated by spraying them with distilled water containing 0.01% Tween^®^ 80. The successful colonization of plant tissue by inoculated EEPF was systematically evaluated after completing each experiment. All leaves that were used in the preference tests were investigated. For plants that were used for VOC collection, once the experiment was completed, the whole upper part of each plant was collected (including stems for tobacco). In every case, samples were rinsed in tap water and treated under sterile conditions based on the method used by Rondot et al. [48]. Samples were surface sterilized separately by soaking them in 0.5% active chlorine (NaOCl) containing 0.01% Tween^®^ 80 for 2 min, followed by 70% ethanol (EtOH) for 2 min. They were then rinsed three times with sterile distilled water and placed on autoclaved filter paper to dry off. About three 1.5 cm^2^ pieces of each leaf were used in the preference tests. Nine pieces of tissue (including two pieces of basal, central, and apical leaves and three pieces of a 1 mm thick cross-sections of tobacco stem that was replaced by an additional leaf for wheat) were collected from plants for use in VOC collection. The pieces from the same plant were grouped together and were first pressed on sterile PDA culture medium in Petri dishes to determine whether any spores were present on their surface. The pieces were then placed on a new culture medium to incubate. The disinfection process was also evaluated by plating three replicates of 100 µL of the last rinse water on three different PDA media. Afterwards, all plates were sealed and placed in darkness in an incubator at 25 ± 1 °C. Ten days later, fungal colonies growing from internal plant tissues were visually examined according to their characteristics: “white dense mycelia, becoming creamy at the edge” for *B. bassiana* [49] and “conidial mass dark yellow-green” for *M. acridum* [50]. When one tissue from a single leaf showed fungal growth, the whole leaf was classified as being endophytically colonized [48]. The results of the preference tests were only validated for endophytically colonized leaves. Plants used for VOC sampling were classified as being endophytically colonized when fungal growth was observed on at least five out of the nine tested tissues. No fungal growth was recorded in any of the rinsed water samples or on the culture media on which plant tissue imprints were marked.

Five days after seedlings were transplanted, five individuals of *M. persicae* or *R. padi* from VP-1 or VP-2 that were 5 days old were confined in a clip cage [51] on the bottom of a single leaf of each tested plant for virus inoculation. The Inoculation Access Period (IAP) lasted 4 days for *M. persicae* on tobacco and 5 days for *R. padi* on wheat [52,53]. Afterwards, insects were removed from the plants with a brush. To exclude any bias related to the virus inoculation on different plant groups, healthy plants for further fungal inoculation and control plants were infested by five non-viruliferous insects. Incubation time lasted 14 days for PLRV on tobacco and 21 days for BYDV-PAV on wheat [52,53]. Virus inoculation was assessed before the preference tests by enzyme-linked immunosorbent assay (ELISA). Two kits were used following the manufacturer’s instructions: Double Antibody Sandwish ELISA (DAS-ELISA) for PLRV on tobacco using a DSMZ kit and Triple Antibody Sandwich ELISA (TAS-ELISA) for BYDV-PAV on wheat using an Agdia kit (Agdia Inc., Elkhart, IN, USA). Samples were collected from the first fully expanded leaf on each plant. A plant was considered as infected if the optical density was at least twice that of the negative control. Only plants that were effectively infected were used.

### 2.5. Design of the Preference Bioassay

Dual-choice tests were implemented for both the MTP and RWB models. The experimental setup was based on the aphid dual-choice arena presented by dos Santos et al. [14], and was adapted according to our plant models (Figure 1). Petri dishes of 9 cm in diameter were used in every case. For the MTP model (Figure 1A), two leaf discs (1.5 cm in diameter) were randomly sampled from the tested plants, and were kept for 10 min in the dark in a box lined with wet filter paper, to allow volatile emissions to be reduced due to injury [54]. The leaf discs were then placed in a dish 4.5 cm apart from each other. Three leaf discs were sampled from three different leaves of a single plant. Leaf discs were renewed for each replicated test. For the RWB model (Figure 1B), two pairs of 1.2 cm oval holes were pierced in the dish; 4.5 cm was left between the holes of one pair, and 5.5 cm was left between each pair of holes. One leaf from each tested plant was carefully introduced to the dish from the first hole, and then exited from the second hole on the same side, providing approximately 2 cm^2^ surface area available to insects. Two leaves were used for each plant.

All of the experiments were conducted under uniform lightening from 16-W cool white fluorescent lights in a climatic room at 22 ± 1 °C and 70% RH. Twenty newly molted adults were released in the center of the arena, which was immediately closed, using the cap of a 1.5 mL Eppendorf tube (VWR, Radnor, PA, USA). The number of aphids on the leaves or leaf discs was recorded 60 min later.

Choice tests were first implemented with only healthy plants (HP) to check for bias in the experimental setup. Pairwise comparisons were subsequently performed among HP, virus-infected plants (VP-1 or VP-2), and plants inoculated with either *B. bassiana* (BP) or *M. acridum* (MP). For each combination, the insects used were either viruliferous or non-viruliferous. For the RWB model, the experiments were also performed with winged and wingless individuals. It was not possible to use winged and wingless individuals for the MTP model, because too few winged individuals emerged on tobacco. Each pairwise comparison between plants was repeated 15–26 and 13–17 times for the MTP and RWB models, respectively.

### 2.6. VOC Sampling and Analysis

Headspace volatiles from the upper parts of seedlings were collected using a dynamic “push–pull” pump system (Benchtop system: CASS6–MVAS6; Volatile Assay Systems^®^, Rensselaer, New York, NY, USA). Each plant treatment was sampled (i.e., HP, VP, BP, and MP) with one blank (pot containing substrate) for each sampling session. Shortly after the sampling period, aerial plant parts were directly excised and weighted to calculate the amount of each VOC under the different plant modalities. After each sampling event, EEPF colonization was verified. Based on the plant model, two different sampling and analysis methods were implemented for VOCs.

*Tobacco model*: Four weeks after transplanting (four-fully expanded leaf stage), VOC sampling from tobacco plants was carried out using the dome and guillotine system, as described by Verheggen et al. [55]. In brief, the aerial part of an individual potted plant was covered by a glass dome (15 cm base-diameter, 15 cm height) placed over a Teflon (Chemours, Wilmington, Delaware) guillotine. All equipment was rinsed using n-hexane >99% (Sigma-Aldrich, St Louis, MO, USA) before each sampling event. The system was set at a constant flow of 350 cc input and 250 cc output. VOCs were trapped for 24 h in a cartridge composed of a thermal desorption tube containing 60 mg Tenax TA^®^. The cartridge was first conditioned at 300 °C for 11 h in a thermal conditioner (TC2, Gertsel, Mülheim an der Ruhr, Germany), and was then placed in one of the air pulling outlets from the dome base. After headspace sampling volatile, all cartridges were stored in a fridge at 4 °C for about 1 week before chromatographic analysis.

Before the analysis of volatiles by GC-MS, 42.5 ng n-butylbenzene (82 ng·µL^−1^) was spiked on each tube. The volatiles were then thermally desorbed using an automatic Thermal Desorber Unit (TD30R, Shimadzu, Kyoto, Japan) set at 280 °C for 8 min. A split ratio of three was used during the injection. Helium was used as the gas carrier with a flow of 1 mL.min^−1^. The cool trap was set at –30 °C. Before injection, the trap was desorbed at 280 °C for 5 min. Samples were then injected into a capillary column (5% phenyl methyl; maximum temperature: 325 °C; length: 30 m; diameter: 250 µm; thickness: 0.25 μm). The temperature program started at 30 °C for 5 min, was then increased by 5 °C·min^−1^ up to 220 °C, and was finally increased by 20 °C.min^−1^ to reach 300 °C. Compounds were identified by comparing their mass spectra with those of the NIST17 database using GCMS Postrun software (GCMSsolution v. 4.50, Shimadzu, Kyoto, Japan).

*Wheat model*: The pots that contained 35 wheat seedlings at the Z16 stage [56] (37 days after sowing) were separately sealed in 4-L glass chambers. The pot was completely wrapped with aluminum foil, to avoid any contamination. The air was cleaned by an activated charcoal filter, and was blown into the glass chamber using a vacuum pump with a constant flow of 650 cc·min^−1^ for 24 h. The Teflon pipe circuit passed through a sampling cartridge (40 mg HayeSep Q, 80/100 mesh; Supelco, Bellefonte, PA, USA) that was placed at the exit of the glass chamber to trap the headspace volatile compounds released by plants. The cartridges were previously cleaned twice by injecting 150 µL n-hexane. The VOCs were eluted to a vial using 200 µL n-hexane. Eighty-six nanograms of n-butylbenzen diluted in n-hexane were added to each sample as an internal standard (IS). Each time, 150 µL n-hexane with 15 µL SI was sampled as a blank. Then, all vials were kept in the freezer at −80 °C before the chromatographic analysis.

Volatile analysis was performed by Gas Chromatography (model 6890) coupled with a Mass Spectrometer system (model 5973) (GC-MS; Agilent Technologies Inc., Santa Clara, CA, USA). An aliquot (1 µL) of each sample was injected in spitless mode. The same column as previously described was used. The temperature program started at 40 °C for 2 min, then increased successively to three following gaps: (1) 4 °C·min^−1^ up to 90 °C; (2) 6 °C·min^−1^ up to 155 °C for 10 min, and finally (3) 25 °C up to 280 °C for 5 min. Compounds were identified by comparing their mass spectra with those of the Wiley 275, and pal 600k databases, using Qualitative analysis navigator (v. B.08.00, MassHunter Workstation Software, Agilent Technologies Inc., Santa Clara, CA, USA).

In each case, a series of n-Alkanes (C7-C30) was injected at the same time to confirm the identification of the library by calculating the retention index (RI). Ri was compared to the theoretical RI from online databases, including PubChem [57], PheroBase [58], and NIST (National Institute of Standards and Technology) [59].

### 2.7. Statistical Analyses

We performed a generalized linear model with a Poisson distribution to test insect preference between treatments, which were pairwise compared: (i) HP, (ii) virus infected plants (VP-1 and VP-2), and (iii) EEPF-inoculated plants (BP and MP). The preference of viruliferous (I+) and non-viruliferous (I-) insects was evaluated based by the number of individuals found on the leaves of the tested plants after 60 min. Factors included plant treatment, the virus infectious status of the insect, and the morphology of the insect (for the RWB model only). This analysis was completed in R version 3.6.1 (R Core Team, 2019).

Volatile profiles from the plant treatments were compared using a permutational multivariate analysis of variance (perMANOVA) with an Euclidian distance matrix and 999 permutations in the R-package “vegan” [60]. Beforehand, the “betadisper” function was used to check the homoscedasticity. Pairwise comparisons were performed when significant differences were detected. The p-values were adjusted using Bonferroni’s method, to avoid type I errors due to multiple analyses. To visualize the spatial distribution of the volatiles collected on different plant treatments, a principal component analysis (PCA) was performed, and plots were generated using the R packages *FactoMineR* [61] and *factoextra* [62]. One-way ANOVA was then used to highlight compounds that were impacted in each plant modality. The average amounts of VOCs collected in each plant modality was compared after checking the normality and homogeneity of variance. Tukey’s pairwise comparisons were computed when significant differences were obtained. Statistical analyses were completed using Minitab software v. 18.1 (Minitab Inc., State College, PA, USA).

## 3. Results

### 3.1. Aphid Preference

No bias was observed in the preliminary test. The theoretical insect distribution of 50% for each tested plant was observed, regardless of the status of insect infection (Appendix A).

*Healthy versus virus-infected plants*: Assays performed between healthy and virus-infected plants (Figure 2) showed a cross-preference depending on the infectious state of insects. Regardless of the insect–plant–virus model, wingless and winged I+ preferred HP, while I- mostly migrated to virus-infected plants. Morphology had no impact on insect preference for the RWB model.

*EEPF-inoculated versus healthy plants:* Wingless *R. padi* and *M. persicae* exhibited a significantly greater response towards EEPF-inoculated plants compared to non-inoculated plants (Figure 3). This preference for BP and MP was significantly stronger when wingless I+ were tested compared to I- in both MTP and RWB models, except for wingless *R. padi* with MP. Compared to wingless individuals, winged *R. padi* were more significantly attracted to BP. In contrast, winged individuals showed no preference between MP and HP. In both cases, the infection state had no significant impact on the choice of winged individuals.

*EEPF-inoculated versus virus-infected plants*: Regardless of EEPF strain and the aphid-virus-plant model, viruliferous insects were significantly attracted to EEPF-inoculated plants, while there was no significant difference for non-viruliferous insects (Figure 4). Winged and wingless *R. padi* showed no significant difference in preference between virus-infected and EEPF-inoculated plants.

### 3.2. Organic Compounds of Volatiles

A total of 22 and 18 compounds, grouped into 11 chemical families, were identified from the various tobacco and wheat plant treatments, respectively (Appendix A). The most abundant compounds were aldehydes (34.22% and 43.76% for tobacco and wheat, respectively), followed successively by terpenes (21.96%), ethers (13.50%), hydrocarbons (12.76%), and ketones (8.07%) for tobacco and by hydrocarbons (18.57%), alcohols (17.98%), and ketones (13.30%) for wheat.

The first two main components of the PCA (PC1 and PC2) represented 50.3% and 64.6% variation in tobacco and wheat, respectively. PCA distinguished clear clusters between treatments (Figure 5). In tobacco, PC1 was mainly correlated to compounds that were specifically associated with VP-1, including hydrocarbons (hexadecane and eicosane), ketone (2-pentadecanone,-6,10,14-trimethyl), diol (4,8,13-duvatriene-1,3-diol) and acetate (3-isopropenyl-2-methylcyclohexyl-acetate). Alcohols (tridecan-1-ol and thunbergol) were also identified on MP. PC2 was correlated to neophytadiene and octadecanal, which were more abundant in HP, and to a group of VOCs associated with BP, including heptanal, nonanal, solanone, and solavetivone. In wheat, PC1 was strongly correlated to glycerol triacetate (triacetin), 2-hydroxydodecanoic acid, 2-propanol-1-chloro-phosphate-(3:1), bis(1-chloro-2-propyl) (3-chloro-1-propyl) phosphate, and hexadecan-1-ol. Most MPs significantly contributed to this axis, except for one primarily composed of decanal and hexadecane. PC2 was mainly correlated to aldehydes (heptanal, octanal, nonanal), ketones (β-Ionone, dihydroactinidolide), and nonadecane (hydocarbon) associated with BP.

perMANOVA showed volatile profiles differed significantly between treatments, regardless of plant model (F_3,19_ = 7.26, *p* < 0.001 and F_3,19_ = 12.88, *p* < 0.001 for tobacco and wheat, respectively). In tobacco, pairwise comparisons confirmed the difference between HP versus the remaining three conditions (see Appendix A for more details). A marginal difference was observed between MP and VP-1 (*p* = 0.06). In wheat, except for HP versus MP (*p* = 0.078) and HP versus VP-2 (*p* = 0.108), all treatments were significantly different.

## 4. Discussion

Choice tests highlighted contrasting behavioral patterns in host-seeking aphids with respect to their infectious status (I+ or I-) in response to different plant modalities (HPs, VPs and EEPF-inoculated). First, the infectious status of aphids clearly influenced their relationship with virus-infected plants: I+ individuals preferred HPs to VPs, while I- individuals preferred VPs to HPs. This finding was consistent with the scientific literature, as it is well-known that insect-borne viruses manipulate their vector [11,13,14,63,64,65,66]. The “Vector Manipulation Hypothesis” is commonly applied to persistently transmitted viruses [67], such as PLRV and BYDV. This hypothesis suggests that the virus influences its vector to move away from already-infected plants, inducing it to spread and feed on new hosts [67]. For instance, Ingwell et al. (2012) demonstrated that virus-free *R. padi* preferred BYDV-infected plants; however, after it acquired BYDV during in vitro feeding, it preferred healthy plants [10]. Similar results were obtained by Rajabaskar et al. (2013) with *M. persicae* and PLRV [11]. Furthermore, EEPF-inoculated plants were more attractive to aphids compared to HPs. This finding was similar to that of Aragon (2016), in which *M. persicae* was attracted to tomato plants inoculated with *B. bassiana* in a multi-choice test [38]. However, the current study is the first to demonstrate that, unlike virus-infected plants, the infectious status of insect vectors does not interfere with their host-seeking behavior in response to EEPF-colonised plants. Interestingly, choice tests performed between virus-infected (both VP-1 and VP-2) and EEPF-inoculated plants showed that most I+ preferred EEPF-inoculated plants, whereas I- exhibited no preference, irrespective of the insect-plant model and microbe strain. This finding was consistent with two previous observations in that: (1) I+ individuals were more attracted to EEPF-inoculated plants, due to the concurrent absence of the virus and presence of endophytes in the latter; and (2) I- individuals had no preference, because both tested modalities previously proved to be attractive to them. Thus, EEPF- and virus-inoculated plants are considered to be of equivalent quality to aphids. Finally, winged *R. padi* showed a similar behavioral pattern to their wingless counterparts. Thus, variation to morphology had no significant influence on their choice. It was not possible to evaluate the effect of morphology on the preference of *M. persicae*, due to the lack of winged forms. This could be caused by the fact that the insect clone used in this study (MpCh4) was a non-tobacco-specialist [68].

How these differences in behavior were mediated remains an open question. Insect-borne viruses appear to regulate the ability of their vectors to locate a host plant by stimulating their olfactory system [69]. Thus, plant volatiles might have significantly influenced our experimental setup. Consequently, the behavioral tests indicated that: (1) the headspace volatiles from HPs acted as a baseline towards which the aphids (whatever their infectious status) were tuned into by default; (2) some featured compounds (or a specific combination of these) that were emitted on infection in VPs had a contrasting effect (repellent to I+, attractive to I-); and (3) EEPF-inoculated plants emitted active compounds that were different to VPs, because both I+ and I- aphids were attracted to them. Interestingly, the headspace volatiles collected from the different plant modalities were qualitatively and quantitatively different. In particular, aldehydes (including heptanal, octanal, nonanal, and decanal) were more abundant in EEPF-inoculated plants, regardless of strain. These compounds are attractive to aphids, including *M. persicae* and *R. padi* [70,71]. Further experiments are required to screen for and accurately validate candidate active compounds that form part of the olfactory signature of each tested modality. Such experiments would include electrophysiological recordings at the antennal level and choice tests with collected or synthetic volatiles.

A major perspective of this study is to investigate whether aphid preferences alter the efficiency of virus transmission in different systems. The fact that for instance aphids were attracted to EEPF-colonized plants does not seem beneficial to the host, potentially decreasing fitness and increasing the spread of virus. Investigating the settlement behavior of sap-feeding insects and transmission dynamics on different plant modalities might help to evaluate the potential benefits conferred by endophytes. The most recent reports by González-Mas et al. (2018 and 2019a) showed that, by colonizing melons, *B. bassiana* altered the feeding behavior of *Aphis gossypii* (Glover) and significantly reduced inoculation rates by 21.9 and 24.4% for *Cucumber mosaic virus* and *Cucurbit aphid-borne yellow virus*, respectively [27,72]. Furthermore, the influence of aphid preferences on their life history and population dynamics, as well as those of their natural enemies, requires investigation. Many studies have already reported the impact of EEPF plant colonization in this regard [26,30,31,39,41,73]. In a multitrophic context, *R. padi* carrying BYDV was more likely to be parasitized by *Aphidius colemani* (Viereck) compared to virus-free individuals [74]. González-Mas et al. (2019b) reported that *A. gossypii* reared on *B. bassiana* inoculated plants were preferentially consumed by their natural enemy *Chrysoperla carnea* (Stephens) over those reared on uninoculated plants [75]. Commercial fungal strains, such as those used in the current study, are commonly used in inundative treatments via foliar application [76,77,78]. Their role as endophytes remains poorly understood. Determining their influence on the aphid-borne virus pathosystem could be crucial for developing integrated pest management strategies.

## 5. Conclusions

Our study confirmed that the infectious status of aphids influences their relationship with virus-infected plants following the “Vector Manipulation Hypothesis”. This phenomenon was not observed when virus-free plants were compared to EEPF-inoculated plants, especially viruliferous and non-viruliferous aphids, which were both attracted to EEPF-inoculated plants. Thus, our study is the first to demonstrate that the infectious status of insect vectors does not interfere with their host-seeking behavior in response to EEPF-colonized plants. Moreover, non-viruliferous insects showed no preference between virus-infected and EEPF-inoculated plants, possibly because both plant modalities were qualitatively equivalent. Finally, volatilome analysis confirmed that the presence of endophytic entomopathogenic fungi in leaf plant tissues altered the profile of volatiles emitted by the latter. Our findings provide an interesting research perspective on how EEPF contribute to the aphid-borne virus pathosystem.

## Figures and Tables

**Figure 1 insects-11-00435-f001:**
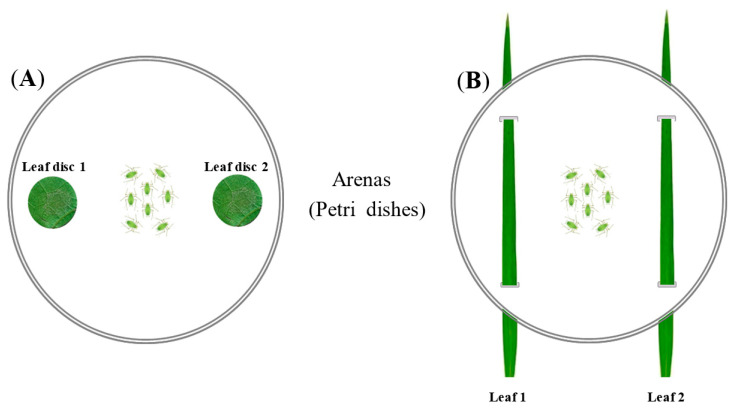
Experimental design for the preference bioassay for Myzus–Tobacco–*Potato Leafroll Virus* (PLRV) model (**A**) and Rhopalosiphum–Wheat–*Barley Yellow Dwarf Virus* (BYDV) model (**B**). Twenty viruliferous (I+) and non-viruliferous (I-) aphids were allowed to choose between the following treatments: healthy plant (HP), virus-infected plants (VP-1 or VP-2) or plants inoculated either with *B. bassiana* (BP) or *M. acridum* (MP). n = 13–26.

**Figure 2 insects-11-00435-f002:**
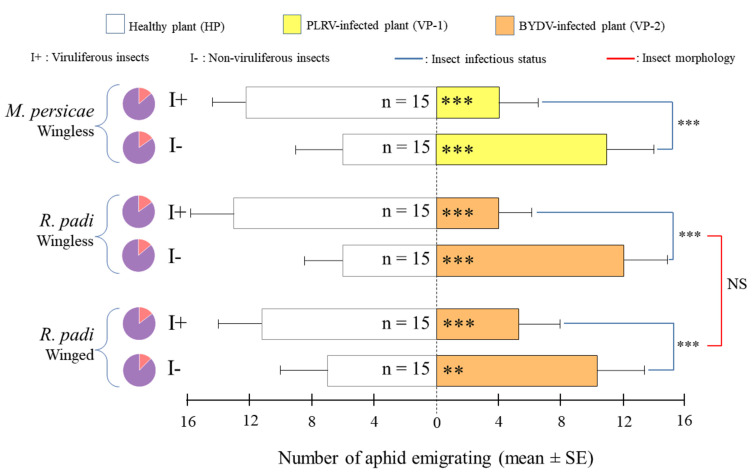
Mean ± SE (standard error) of insects migrating on healthy (HP) and virus-infected (VP-1 or VP-2) plants pairwise-compared during dual-choices tests. Each test was performed by releasing 20 viruliferous (I+) or non-viruliferous (I-) insects and observed after 60 min. Pie charts denote the proportion of responding (purple) versus non-responding individuals (orange). n: number of replicates; *, **, *** and NS for *p* ≤ 0.05, *p* ≤ 0.01, *p* ≤ 0.001 and not significant at α = 0.05, respectively.

**Figure 3 insects-11-00435-f003:**
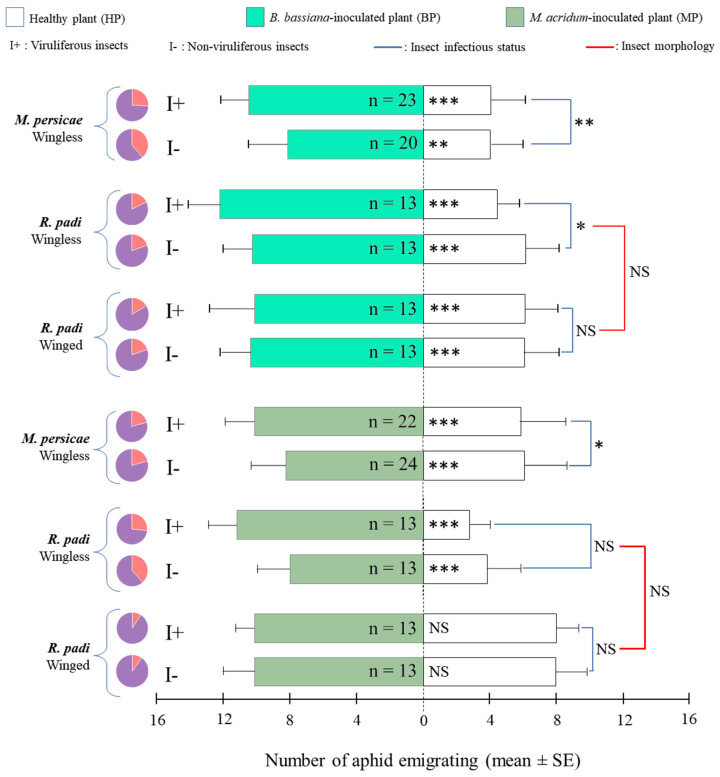
Mean ± SE of insects migrating on healthy (HP) and EEPF-inoculated (MP and BP) plants pairwise-compared during dual-choices tests. Each test was performed by releasing 20 viruliferous (I+) or non-viruliferous (I-) insects and observed after 60 min. Pie charts denote the proportion of responding (purple) versus non-responding individuals (orange). n: number of replicates; *, **, *** and NS for *p* ≤ 0.05, *p* ≤ 0.01, *p* ≤ 0.001 and not significant at α = 0.05, respectively.

**Figure 4 insects-11-00435-f004:**
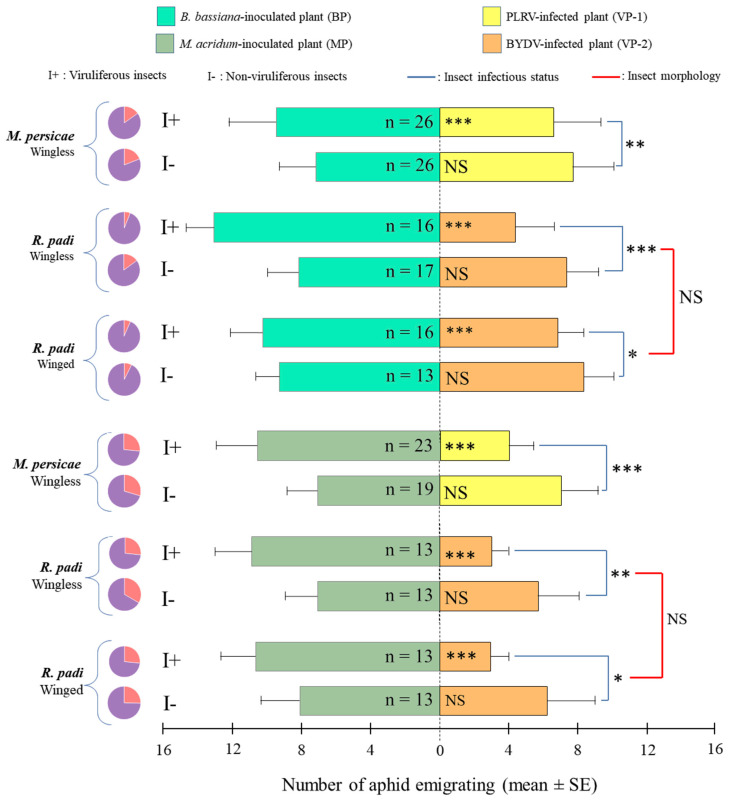
Mean ± SE of insects migrating toward virus-infected (VP-1 or VP-2) or endophytic entomopathogenic fungi-inoculated (BP or MP) plants pairwise-compared during dual-choices. Each test was performed by releasing 20 viruliferous (I+) or non-viruliferous (I-) insects and observed after 60 min. Pie charts denote the proportion of responding (purple) versus non-responding individuals (orange). n: number of replicates; *, **, *** and NS for *p* ≤ 0.05, *p* ≤ 0.01, *p* ≤ 0.001 and not significant at α = 0.05, respectively.

**Figure 5 insects-11-00435-f005:**
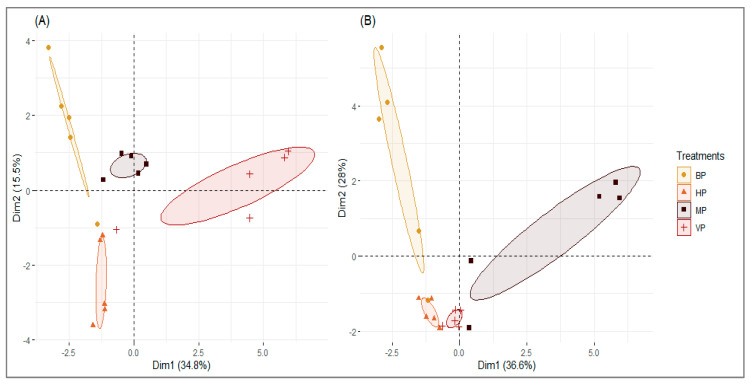
Principal component analysis (PCA) of the volatile data collected during 24 h from tobacco (**A**) and wheat (**B**). *B. bassiana*-inoculated plants (BP); *M. acridum*-inoculated plants (MP); virus-infected plants (either VP-1 or VP-2); healthy plants (HP); n = 5 replicates.

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
