# Peer review of "Does the Infectious Status of Aphids Influence Their Preference Towards Healthy, Virus-Infected and Endophytically Colonized Plants?"

_insects, 2020, doi:10.3390/insects11070435_

Round 1
Reviewer 1 Report
Insects MS 813581 Does the infectious status of aphids influence their preference towards healthy, virus-infected and endophtyically colonized plants? Mabola et al.
In this interesting paper the authors looked at two aphid species, Myzus persicae and Rhopalosiphum padi to see if the infectious status of their respective host plants, tobacco and wheat, changed the attractiveness of them to the aphids. They also looked at whether viruliferous and non-viruliferous aphids varied in their responses. They showed significant behavioural differences between infected and non-infected aphids and also an interaction between the type of infection that the plants were suffering from.
In the main, the writing was clear and the analysis appropriate. I would, however, caution the authors from beginning sentences with the word however, however tempting it might be. It si also more grammatically correct to compare with rather than to.
Line 14 damage not damages
Line 35 It is more correct to say sap feeding – aphids do not suck phloem, which is their main food source. Phloem is under pressure so in order to prevent being blown up, they regulate the flow of phloem using a tracheal valve. They do, however, on the less frequent occurrences when they feed on the xylem, suck, which is why sap-feeding is a more accurate description.
Line 144 Not everyone knows what a clip cage is – you need to reference it e.g. MacGillivray, M.E. & Anderson, G.B. (1957) Three useful insect cages. Canadian Entomologist, 89, 43-46.
Line 161 Truly random, i.e. using random number generator or do you mean haphazardly?
Line 171 Newly moulted adults?
Line 215 Use a growth stage descriptor e.g. Zadoks, J.C., Chang, T.T. & Konzak, C.F. (1974) A decimal code for the growth stages of cereals. Weed Research, 14;, 415-421.
Author Response
Sir, Madam,
Thank you for your constructive comments that helped us improve this paper. Please find the answers to your questions on the attached document.
Yours sincerely

Reviewer 2 Report
In this paper authors looked to see if preferences of aphids varied according to their infectious status and if viruliferous and non- viruliferous aphids could distinguish between healthy and infected plants as well as between EEPFs colonized and healthy or infected plants. The study system chosen consisted of two aphid species belonging (R. padi and M. persicae), two plant hosts (N. tabacum and t. estivum), two virus species, PLRV and BYDV and two EEPFs (B. bassiana and M. acridum).
The paper is overall informative, and the experiments well conducted and interpreted. One major criticism is that authors could have incubated and tested the leaf disks exposed to viruliferous aphids and determined their infectious status, to understand if aphids were able to transmit the viruses, and if transmission were correlated with choice. Is in fact not clear how the preferences shown by aphids in this study could influence virus transmission dynamics. These limitations should at least be part of the discussion.
In the abstract remove their hosts since both aphids are generalists.
Describe the type of transmission and relationship between the two viruses and their 2 vectors.
Is N. tobacum a good host for M. persicae? In L. 184 the authors seem to agree that this is not a good host for aphids, since aphids could not complete their life cycle on tobacco plants. Add in the discussion if this is the case, and the relevance of the two systems used, in terms of research purposes or in terms of real-life application.
If known, describe how common the colonization of the two hosts by the two EEPFs occurs in nature, and how stable it is, when the two EEPF are exogenously supplied.
A further revision of the manuscript for English would be beneficial.
56: ‘this promotes’ or ‘these promote’? (referred to the list before or just to the last item in the list?)
59: substitute ‘Regarding all these information’ with something more meaningful, such as ‘In support of this line of research’ or ‘To contribute to this knowledge’. After this sentence, explain why it is important to do this. For instance, if this knowledge can be potentially used for aphid/virus management.
L.92: substitute ‘by doing so’ with ‘via insect transmission’
99-104: Were the identities of the EEPFs verified in any way?
113: reached?
117: defined the suspension (water? Water +Tween?)
118: ‘per plant’ and not ‘by plant’
121: how was the colonization evaluated?
122-124: At what time point were samples taken from the plants used for VOC?
127: ‘3 pieces were collected’ and not ‘were collected 3 pieces’. This sentence is not very clear overall and should be rephrased.
136-137: was any other molecular method used for EEPFs identification, aside from presence of mycelia and conidia?
L.190: was each treatment coupled with one blank, or was the blank used in a separate collection?
193: not ‘verify’ but ‘verified’
195: ‘transplant’ instead of ‘transplantation’
L.199: ‘rinsed’ instead of ‘washed’. Define the purity and brand of n-hexen here instead of below L. 221-222
200: ‘in a thermal’, instead of ‘thanks to’
201: define ‘They’: the tubes? The membranes?
202: ‘air’ instead of ‘airs’
205: Prior to
209: remove ‘;’ and replace with ‘,’
216: ‘chambers’
218: per min: use consistency in the annotations (per min vs. min-1)
219: a sampling
234: was injected
235: ‘ was compared’ instead of ‘them compare’
241: ‘throughout’ or ‘by’?
Figure 1: add to the figure legend what were the treatments tested, the number of aphids released, and the number of tests performed, to make the legend more self-explanatory.
Figure 4: maybe infected aphids are repelled by infected plants?
Section 3.1. It is not completely clear if the results reported are due to differences in the quantity of volatiles emitted by plants in the different treatments or are due to different compounds. Can the authors be more specific to this regard? This seems especially important when in the discussion the authors report that ‘. Interestingly, the headspace volatiles collected from the different plant modalities were qualitatively and quantitatively different. Especially, aldehydes (including heptanal, octanal, nonanal and decanal) were more abundant in EEPF-inoculated plants, regardless of the strain’.
337: consider changing ‘to spread and conquer new hosts’ with ‘to feed on new hosts’
346: consider removing ‘double’
Discussion: add a paragraph discussing how the choice tests performed here can impact virus spread. These tests indicate a preference for the vectors, but do not offer much indication regarding the feeding propensity of the aphids on the different plants, and if these choice result in changes in transmission efficiencies in the different systems. Host manipulation, as reported here, is hypothesized to favor pathogen transmission, and this is an important result missing from this study
Author Response
Sir, Madam,
Thank you for your constructive comments that helped us improve this paper. Please find the answers to your questions on the attached document.
Kind regards

Round 2
Reviewer 2 Report
Thanks for addressing the comments and requests for this report